# Exploring the Benefits, Barriers and Improvement Opportunities in Implementing Automated Dispensing Cabinets: A Qualitative Study

**DOI:** 10.3390/pharmacy13010012

**Published:** 2025-01-29

**Authors:** Abbas Al Mutair, Alya Elgamri, Kawther Taleb, Batool Mohammed Alhassan, Mohamed Alsalim, Horia Alduriahem, Chandni Saha, Kawthar Alsaleh

**Affiliations:** 1Almoosa Specialist Hospital, Al-Ahsa 36342, Saudi Arabia; 2Nursing Department, Almoosa College of Health Sciences, Al-Ahsa 36342, Saudi Arabia; 3Department of Medical and Surgical Nursing, College of Nursing, Princess Nourah bint Abdulrahman University, Riyadh 84428, Saudi Arabia; 4School of Nursing, University of Wollongong, Wollongong, NSW 2522, Australia; 5Prince Sultan Military College of Health Sciences, Dhahran 31932, Saudi Arabia; 6Faculty of Dentistry, University of Khartoum, Khartoum 11111, Sudan

**Keywords:** automated dispensing cabinets, ADC, benefits, barriers, improvement opportunities

## Abstract

Technology has increasingly influenced the provision of healthcare services by enhancing patient safety, optimising workflows, and improving efficiency. Large healthcare facilities have adopted automated dispensing cabinets (ADCs) as an advanced technological solution. A key gap exists in understanding the ADC implementation experience in different contexts. Therefore, this study seeks to fill this literature gap by exploring key stakeholders’ perspectives on the benefits, barriers, and improvement opportunities related to ADCs, offering valuable insights to support their effective integration across various healthcare settings. This qualitative study was conducted in Saudi Arabia. The implementation of ADCs generally has positive outcomes for all staff. The system has brought about enhanced medication tracking, greater time efficiency, along with reduced workload and medication errors. However, there are barriers to their implementation, including changes in workflow and workload distribution, cabinet design, technical medication management challenges, and the need for staff training. To maximise the effectiveness of ADCs, healthcare organisations should focus on improving operational workflows, providing ongoing staff training, and maintaining robust system monitoring. Additionally, manufacturers should focus on advancing technology to further enhance the efficiency and functionality of ADCs.

## 1. Introduction

Technology has increasingly influenced the provision of healthcare services by enhancing patient safety and care quality, streamlining processes, and improving efficiency [1,2]. Automated dispensing cabinets (ADCs) are one such technology that has been adopted in large healthcare facilities. These systems integrate electronic prescribing and dispensing to enhance medication management [3]. Integrating ADCs into hospital workflows often requires changes in procedures and training. While this can initially disrupt existing practices, it ultimately leads to better and more organised medication management processes and nursing workflow [4,5]. The core benefit of these systems lies in their ability to significantly reduce medication errors and prevent delays in medication administration [3,6,7]. By streamlining the dispensing process, these systems enhance efficiency, minimise human error, and ensure timely access to medications, all of which contribute to improved patient safety and overall quality of care. Furthermore, ADCs help healthcare staff manage inventory more effectively and maintain accurate records, which supports better clinical outcomes [3,8,9]. Another notable advantage of ADCs is their ability to significantly decrease the incidence of returned items, breakages, and medication losses [8]. The implementation of ADCs, however, is not without its challenges. Numerous factors, including issues regarding medication stock, barcodes, cabinet design, waiting time, staffing, and workforce, can impact the success of these implementations [5,9].

Saudi Arabia is undertaking a comprehensive transformation of the healthcare system to develop an advanced and sustainable model that aligns with the strategic objectives of Saudi Vision 2030 and the National Transformation Program. A key aspect of this transformation is the emphasis on improving medication management practices to enhance patient safety and operational efficiency [10]. Hospital pharmacists play a pivotal role in this process by implementing evidence-based practices that ensure the safe and effective delivery of medications. As part of these initiatives, the integration of innovative technologies such as ADCs has gained prominence. ADCs offer the advantage of being fully computer-operated and closely monitored systems [11]. They manage the storage, distribution, and tracking of medications directly at the point of care within the wards, ensuring easy access to medications near the site of use while enhancing efficiency and accountability [3]. With ADCs, once the physician enters the prescription into the electronic medical record (EMR) system, the medication should first be reviewed and evaluated for safety by a pharmacist prior to the medication being automatically authorised for dispensing [12]. Nevertheless, in urgent situations and based on verbal orders, nurses may use the override function to access medications immediately. Nurses can then retrieve the required medication directly from the cabinet, ensuring faster delivery, reduced errors, and improved inventory management. Unlike traditional methods, manual floor stock systems or medication carts that store a 24-h supply of medications in individual patient-specific containers, ADCs represent a significant advancement in medication management practices [12,13].

Despite the use of ADCs in the institution where this study was conducted has generally been viewed positively, barriers still exist, 8.4% of healthcare professionals have previously reported that the use of ADCs is time-consuming and more difficult, highlighting the need to explore the current practice and use of such systems [14]. A systematic review that investigated the impact of ADCs on the clinical and economic aspects cautioned that any positive outcomes might be unique and institution-specific, with success relying on how well ADCs are integrated with the process of medication distribution at each setting [15]. A key gap exists in understanding the ADC implementation experience in different contexts. Therefore, this study seeks to fill this literature gap by exploring key stakeholders’ perspectives on the benefits, barriers, and improvement opportunities related to ADCs, offering valuable insights to support their effective integration across various healthcare environments.

## 2. Materials and Methods

### 2.1. Study Design

In this study, a qualitative research design was used to explore the benefits and barriers related to implementing ADCs at Almoosa Specialist Hospital (ASH), a leading provider of healthcare services. ASH, which has a capacity of 460 beds, is the first and largest private healthcare institution in AlAhsa, Saudi Arabia. The integration of ADCs within ASH was introduced in September 2021, as part of an initiative to enhance medication management efficiency and patient safety. Currently, the hospital has deployed 25 functional ADCs across 20 units. Prior to ADC implementation, the hospital was using both manual floor stock systems and medication carts for storing a 24-h medication supply with patient-specific containers.

### 2.2. Population and Sampling

Purposive sampling was utilised to ensure the involvement of key stakeholders in the implementation process, including pharmacy staff, medical informatics pharmacists, and nurse managers. The summary of the participants’ characteristics is presented in Table 1.

### 2.3. Data Collection

This qualitative study involved the use of semi-structured interviews to collect the data. The interviews were held in a meeting room at the hospital. Each interview took about 30 to 40 min. All interviews were attended by two investigators, A.A.M. and K.T. The interviewers were the research center director and the pharmacy supervisor. The interview guide was prepared ahead of time and was the same for all interviews. The questions were open-ended to allow detailed answers. At the beginning of the interviews, the interviewer focused on building rapport by greeting the participants and asking about the nature of work, including working unit, job title, and duration of experience with the ADCs in the hospital, along with educational background. This approach was intended to create a comfortable environment for the discussion. The interviews were structured using a guide that focused on key topics related to user experience, challenges, and the perceived benefits of the ADCs. The data collection process proceeded till data saturation was attained. Interviews were audio-recorded and then transcribed into written texts.

### 2.4. Thematic Analysis

The recorded interviews were transcribed to facilitate comprehensive qualitative analysis. Each transcription was carefully reviewed for accuracy, ensuring that all verbal nuances and context-specific details were captured. The transcriptions were then analysed thematically to identify patterns, themes, and insights pertinent to the study’s objectives. Thematic analysis followed that practical guide by Braun & Clarke [16].

## 3. Results

The benefits, barriers, and improvement opportunities in implementing ADC were explored through semi-structured interviews. This paper shows how the participants reflected on their ADC experience. The results are presented in three themes and several subthemes, as shown in Figure 1 and Table 2.

### 3.1. Theme 1: Benefits

The implementation of ADCs has had a positive impact on all staff in general. By automating medication preparation and dispensing, the system has led to increased time efficiency and reduced effort, improved medication management, better tracking and monitoring, and reduced stress and medication errors. Nurses gained benefits, including easier access to medications and reduced medication errors. Inventory management has also been streamlined through the use of ADC, eliminating the need for manual inventory counts.

#### 3.1.1. Time Efficiency and Reduced Workload

Participant 4 emphasised the benefit of having ADC and how it saves time compared to manual work:

“The hospital work for nurses has decreased. Before, when we had only floor stock medications, I would spend two hours per day when it is the night shift to request medications. You will count how many were left and you will request, it will take time, it will take two hours, now no more.”

Participant 7 agreed that it is very useful, saying:

“Of course, it will help us. It will support us. Definitely, it’s helping since most of the medications we need are inside the ADC. So, definitely, it’s helping.”

Participant 8 highlighted the benefits of ADC by reflecting on his recent experience with another unit that does not operate with ADC. He said:

“There is no ADC. It is a disaster! Every morning, I find extra medication. It gives me a workload, I have to investigate, I have to track all these.”

Furthermore, when discussing the benefits of ADCs, Participant 1 feels that ADCs benefit and decrease the work for nurses and pharmacists equally, saying:

“This is my feeling that it is for both. Because the pharmacists need to prepare and to wait for the trolley and to, uh, too much work, and I think, uh, for the nurses to double-check the high alert medication. And you know, these tasks were very hard for the pharmacy staff and the nurses.”

Participant 10 also noted that a major benefit of ADCs is that they simplify the process of requesting new medications and reduce dispensing wait times, which in turn increases patient satisfaction. She said:

“It’s easy. If you have a new medication that has not been activated. So, no need for you to wait 30 min or an hour for the pharmacy to dispense what you have on the ADC. And it’s also satisfying for the patients because it speeds up the process of administering their medication. Just put their badge number, scan, and go to the other room and give them the medication.”

Participant 4, who is a charge nurse in the ER, thinks that ADC is very useful to them. She said:

“With ADC, it is more automated and it’s really helpful for us because there is no need to request, no need to charge, it’s all automated.”

The benefits of ADCs are also well appreciated after initial training, as pointed out by Participant 5, who explained:

“Yes, after training, it becomes very easy. When it came to the first time, it was strange for them. I tell them that it’s very helpful. It will save you time, and you will adapt to it. Once you become familiar with it, it becomes easy. They do not need that much training. Just a few steps. It’s easy for them, and then they like it. Really, for most of the staff, when they start to work with ADC, they appreciate the idea.”

Participant 2, who is a pharmacist had different experiences working with ADC. As the duties alternate between shifts, Participant 2 described how ADC reduces the physical efforts when being an inpatient pharmacist, saying:

“After having ADCs in our hospital, being an inpatient pharmacist is the best thing for every pharmacist. Being there is the best.”

#### 3.1.2. Reduced Medication Errors

Participant 3 explained how ADC provides an important feature that decreases medication errors. He said:

“The stock is more regulated nowadays, and also, by using the ADC, you are sure that you are giving the correct dose to the patient. Because you know, after the doctor orders, the pharmacy will verify it. So, with this flow, the correct dose is given to the patient, there are fewer medication errors, and there is more revenue for us in the hospital ER.”

Furthermore, Participant 6 noted that the system’s features help prevent errors by showing precise quantities that are needed and alerting users if the dosage exceeds the requirement. However, an error might still occur. Participant 6 stated:

“Even the system will show you exactly how many tablets, vials, or ampules you need to take. It will alert you that if you have like, consider the dose is five milligrams and a single tablet is ten milligrams, it will show you that the tablets have exceeded the dose you need, and it will give you an alert. But as a person with no ADC experience, a nurse might administer the full tablet without looking that it’s half of the dose supposed to be given.”

Participant 9 highlighted that the main advantage of ADC is that it gives nurses some relief from the stress associated with medication errors:

“I have to say that having technology alleviates some of the stress of having a medication error. As a nurse, this is a major mistake—administering the wrong medication. Dealing with high-alert medication is like a nightmare for any nurse. If she is not focused, or something distracted her, she would have an error that can risk the patient’s life.”

Participant 6 also said:

“Working with ADC really maintains a safe practice and reduces medication errors.”

#### 3.1.3. Enhanced Medication Tracking

In discussing the tracking and ADC reports, participants primarily focused on the benefits and frequency. All participants agreed that the reports are highly advantageous, aiding in workload reduction, discrepancy detection, and monitoring of human behaviour. Regarding frequency, participants noted that they either received sufficient or even an excess of reports.

Participant 3 explained how ADC provides an important feature that allows them to track staff behaviour for any errors. He said:

“It’s important to track the staff’s behaviour. We should trust our nurses, but sometimes we should have a system to track them properly to identify any errors.”

Participant 3 also noted that receiving reports from the pharmacy reduced the workload significantly:

“Honestly, the pharmacists are doing a great job, every day the pharmacists send us the report of the stock discrepancies and we use this report to correct them immediately. So, we do not need to go for a stock adjustment every time.”

Participant 6 discussed the reports’ functionality:

“I need the report, I have it on my system, I can track it, I can export, I can make it as Excel and I can track any medication, even as a group of medications, or individual one. I can go one by one.”

In general, all participants agreed that they either had sufficient reports or more than enough. None of the participants reported a need for additional reports. However, all agreed that the error notifications can be overwhelming due to their volume. Participant 3 noted:

“The problem is that we receive this message from all departments concerning their ADCs. So, because I am receiving messages from every department, sometimes I might overlook them, thinking they are not meant for me.”

Participant 6 also raised this concern:

“We are receiving this error notification for all the departments or for any violations. We can consider it another challenge because I receive many error notification messages.”

Nevertheless, the participants had different practices to overcome the issue regarding how often they check reports. Participant 3 stated:

“I create an e-mail folder specifically for them. So, I open them, let’s say, twice or three times per shift to look over the ER error notifications.”

Participant 4 reported checking emails immediately:

“When I am on duty, we check every email that is sent we check them all.”

Participant 6 mentioned checking reports every few days:

“I check them every 3 or 4 days, to be honest, or as needed. Checking daily would be very difficult. I mean, we have a lot of other work to do.”

Participant 10 stated:

“As much as I can. As long as I can.”

When the participants were asked if they needed any extra reports or notifications, all of them said they did not. Participant 3 said:

“I don’t need more than this report to check the behaviour of the staff and to ensure that what is in the system and the actual are the same.”

Participant 9 also thinks she has enough notifications. She said:

“I get notified when my stock is going low; I get notified when there is staff behaviour, I get notified when there are missing charges, I always get notified, so I’m being notified enough, I guess.”

### 3.2. Theme 2: Barriers

Implementing ADCs in the hospital has introduced several new challenges and barriers. These challenges encompass system integration and reliability, medication management, and cabinet design, as well as changes in workflow and overload distribution. Below are the various subthemes and the specific issues highlighted by participants.

#### 3.2.1. Staff Training

Effective integration of ADCs with existing hospital systems is crucial for their successful implementation. However, participants reported several integration challenges. Participant 1 described the extensive discussions and training required to shift from a manual to an automated medication flow. She noted:

“It took a lot of discussion because we wanted to understand how we will successfully implement every message. This was the main challenge. And after that, how about the training? And how we will change the manual flow to an automated flow?… And how will we train both the pharmacy team and the nurses? It was very, uh, huge work.”

Several participants emphasised the paramount importance of training for the effective use of the ADC system. For instance, Participant 4 underscored the necessity of training for any new process introduced to an organisation:

“For each process, you need training. For a new process that will be introduced to any organization, you must follow the guidelines communicated during the training. Without the training, you cannot do it.”

This sentiment was echoed by others who agreed that no employee in their department has ever been required to use ADC without prior training. Participant 6 highlighted the role of training in minimising errors, stating:

“To maintain a safe practice for our patient and our nurses as well, we should be trained, I mean, without the training, without knowledge, nobody can perform or achieve a good outcome.”

Participant 2 expressed satisfaction with the training, noting the value of receiving instruction from multiple sources:

“I have been trained by so many people. That was good because everyone at that time had different ideas. So, I got very good training because I got that experience.”

The training was delivered either directly by the ADC vendor or by the MID team within the hospital. Participant 1 explained that the biomedical technician and the MID team initially received training from the vendor:

“The biomedical technician took training from the vendor itself. Because any hardware issue, they need to handle it. So, the biomedical technician met the vendor, the ADC team, and had a training.”

Participant 4 added that initial training was conducted by the vendor, followed by subsequent sessions led by the MID team. Also, she described her experience with the training, which included both vendor-led sessions and internal sessions:

“The first training was with the company, then the following few days were with the MID… It was in English by the vendor themselves when they installed it. I attended many sessions whenever there was a staff who needed to attend, I attended with them… I received training about how to dispense. And once I became a nurse manager, I received specific training from the medical informatics department (MID) in our facility on how I can control the access of the other staff.”

Participant 4 discussed the transfer of training responsibility:

“Yes, for the first training was done by the MID and then we nominated super users in the emergency department. Then the super users also trained the staff. The preceptor will teach the newcomers how to use this. And once you are competent, we can give the pass.”

Participant 1, who was also responsible for conducting training for hospital staff, noted when asked about the flexibility of the training duration:

“It depends on the nurses. Some nurses need only one session to understand the flow. We left them the freedom of choice. One nurse can attend more than two or three sessions.”

Participant 3 supported this view, noting that training was customised to meet individual needs:

“It took us days, but for every day, it was one or two hours, maybe three sessions. Some of our colleagues were able to handle it properly from the first session.”

Although initial training was generally considered adequate, the need for ongoing monitoring, training, and support was highlighted as critical for maximising the system’s potential. Participant 3 explained:

“A few months ago, there were almost huge stock discrepancies in the system because of three or four staff. At the same time, they had repeated actions and unacceptable behaviors. So, we called them for re-training.”

Participant 1 commented on the same issue:

“Based on the transaction errors that we found and the report that we generated from the ADC, there was misuse of the ADC. So, we start to train more and more to decrease these errors.”

Participant 6 also emphasised the importance of ongoing training, and that initial training is often sufficient to start with, but gaps in knowledge usually appear after trainees begin practicing independently. Updates to the system will also require further training, as addressed:

“Two months back I got a group of 18 staff who had behavioral issues and misuse of ADC. I gathered them all, and I trained them everything… Most of the [training] gaps become evident after they start working independently with ADC, we are calling them, and I will train them again for a specific transaction… Now, if there is any update or anything here, we need to train.”

Overall, effective training is crucial for the successful adoption and utilisation of ADCs. All participants highlighted the importance of comprehensive training programs that equip staff with the necessary knowledge and skills to operate the system confidently. Furthermore, they emphasised the importance of continuous on-demand retraining.

#### 3.2.2. Technical and Medication Management Issues

Technical issues, including network problems and system downtime, were reported to impact medication management and availability along with staff efficiency. Medication management within ADCs encompasses several crucial aspects, including medication charging, dosage calculations, and the management of medication batches.

Participant 2 highlighted an incident where discrepancies arose due to integration problems with the Enterprise Resource Planning (ERP) system leading to medication refills not being reflected in the other system:

“It was an integration issue with the ERP We were refilling the medication, and it was not integrated. So, we were refilling and there’s nothing in the system. So, we stopped the refill.”

When Participant 4 was asked about the biggest challenge facing them with ADCs, she said:

“Network and integration issues, especially during downtime.”

One recurring issue is medication charging inaccuracies. Participants reported discrepancies between medications dispensed and those charged to patient accounts. Participant 4 highlighted a system glitch affecting billing, explaining that:

“Some of the medications were correctly taken from the ADC but were not reflected in the EMR. You know, in the ER, fast-moving patients will come, are treated, and then they will be discharged, and their bills must be reflected. When it’s not reflected in the EMR, the receptionist cannot bill them.”

Similarly, Participant 5 reported having error notifications, leading to their medications not being charged despite the correct processes being followed:

“I have an issue with the error notification. It came as not charged; why? When I check the process, everything is okay. Why it’s having error notification, I don’t know.”

Participant 9 reported issues related to sudden system shutdowns and integration problems:

“We have different occurrences when it’s shut down all of a sudden, and sometimes for unknown reasons; it will just hang and shut down when you need medications, and that’s quite challenging till you contact the IT.”

Additionally, Participant 4 noted instances of double charging during system downtimes:

“During the downtime, the ADC will not be charging, but later on, once the network is restored or the downtime ends, it will result in double charging, so you now have a misbalance with your stock.”

In contrast, Participant 3 attributed charging issues to human behaviour rather than system problems:

“The charging issues, sometimes it’s a system problem, but it’s usually a behavioral issue, and you cannot prevent it unless you train the staff and observe the staff.”

Issues related to managing medication batches and expiry dates were also noted. Participant 1 described the challenge of mixed batches in the same bin:

“We start to refill the medication with a lot of batches. Mixed batches in the same bin. We struggled that nurses do not know what batch that they need to take.”

Concerns about medication variance and doses were raised by Participant 5, who discussed the focus on neonatal doses and managing medication volumes:

“In the neonatal, we are focusing on the doses by milliliter. You know, if the antibiotic or medication is prepared by pharmacy, they will give you the exact dose and volume.”

Finally, the use of overridable medications also posed risks and challenges. Participant 3 addressed the safety concerns of overridable medications in the ER, advocating for non-overridable options:

“To be honest, for nurses in the ER to have overridable cabinets, it is easier for them. So, once they have a verbal order from the physician, they will immediately implement the order. But we had an incident where one nurse wanted to administer Atrovent, but he took Atropine from the ADC because it’s overridable. He just wrote ATRO… and he didn’t continue, then he pressed Atropine, and he took Atropine. Thank God it was inhaler therapy, and no major issue happened for the patient. So, it’s very risky when we talk about overridable, risky areas, especially in the ER.”

#### 3.2.3. Cabinet Design

The design of ADCs significantly impacts their effectiveness and safety. The free zone and drawer design have been noted as both advantageous and challenging. Participant 2 highlighted the benefits of these features for accommodating larger vials, stating:

“Well, I think the free zone is helping us and the bins as well, the closed drawers. Both are helpful, to be honest, because we do, like, if we are to refill Paracetamol, it’s a large vial, it’s a glass vial, how can we fit it inside the secured bin.”

Participant 5 highlighted a significant challenge with the design of the free zone:

“I have an issue, a big issue that I noticed from the staff, that all the free zones are near to each other. The light near to this one. Do you know what has happened? The staff took another medication. If you check the ADC, all the free zones are beside each other. Sometimes really, I doubt myself.”

Participant 6 further described the light issue, noting that familiarity with the system is required:

“In the beginning, it was a little confusing for the light because the light or the drawer, when it’s opened, will show you the light itself and this light, it’s coming between two drawers. But after sometimes you need to become familiar, you will consider this line for the light, and the other line for the drawer. After that, you can go easy.”

Despite these issues, several participants emphasised that many problems with cabinet design stem from human behaviour rather than the design itself. Participant 10 noted that compliance issues, rather than design flaws, were often at the root of the problems:

“It will be based on the staff compliance. If they receive medication, just like the medications that were being sent from the pharmacy, they should check it.”

#### 3.2.4. Workflow and Workload Distribution

The introduction of ADCs has significantly transformed medication management processes, yet it has also introduced additional burdens. While nurses have reported a reduction in workload as a benefit and reported easier medication access and fewer errors, the administrative tasks associated with ADCs have increased for other staff members, particularly pharmacy and nursing managers. Participant 2 which is a pharmacist, highlighted the complexities of the refilling process, including medication preparation, checking, and delivery, noting that it is time-consuming and led to a change in the workload distribution:

“The refilling process is hard, it’s not easy. So, we are taking almost one hour… one hour and a half every morning to check the medication we need to refill. Initially, we thought that the workload would decrease for us as pharmacists, but it turned out that it only led to a change in the workflow. The work became heavier for the pharmacists responsible for stocking the cabinets, and decrease for the other pharmacist and nurses.”

#### 3.2.5. Stock Adjustment

System stock adjustment and tracking medication discrepancies were also identified as significant challenges. Participant 6 discussed the difficulty of identifying discrepancies between the system and physical counts, stating:

“The challenges. Mainly we are speaking about variations, the variations on the system compared to the ADC.”

Participant 5 also noted that system stock adjustment is particularly burdensome on busy days, saying:

“I focus on the medications that are used more frequently. If the medication is used within this week, I’ll count it. If medication is not used, I will not. But now, when I’m busy, no. Sometimes I just really can’t keep up and have to stop.”

Participant 6 also noted that stock adjustment adds to the workload, saying:

“The system stock adjustment, here I have an issue, I mean, I don’t always have time to do a system stock adjustment.”

Participant 2 further described the burden of managing both physical and administrative aspects of medication management, saying:

“Because it’s a heavy task. You are not going out of the pharmacy alone; you are carrying a heavy bag, heavy medications, heavy work.”

Participant 6 differentiated the workload between nurses and managerial levels, explaining:

“If I’m speaking about the managerial level, there is a difficulty in the coding of ADC system and these things. Because each type of error shows different codes, but for the staff nurses, no, it’s easy. If I am a staff nurse and I need to dispense the medication from the ADC, it will be easy for me, I will just go directly to the ADC, enter the MRN, I will see which is the ordered medication and I will dispense directly.”

Conversely, Participant 4 believed that stock adjustment itself is not problematic and does not add significantly to the workload, but checking transactions daily is time-consuming:

“To do the system stock adjustment for the discrepancy for medication? No, isn’t an issue, but to check the transaction daily Yes, it is time-consuming.”

Overall, while ADCs have streamlined many aspects of medication management, they have also introduced new challenges that require ongoing attention and adaptation.

### 3.3. Theme 3: Improvement Opportunities

#### 3.3.1. Discharge Medications

Participants also made several suggestions for improvement, such as adding discharge medications to the ADC to expedite discharge processes. Participant 5 said:

“In NICU, the discharge medication is only vitamin D, we rarely have another medication. Sometimes it will cause a delay of discharge because we are waiting for vitamin D. And if we can add the discharge medication and make it chargeable.”

#### 3.3.2. Department-Specific Error Notifications

The participants provided several insightful suggestions for improving ADCs. To begin with, all participants believe that having department-specific error notifications would streamline issue resolution, Participant 3 stated:

“If we can identify it for emergency or a specific department, it will be much better.”

Participant 6 also highlighted this concern:

“If we can identify them separately, department by department, I mean like ICU alone. ER alone. LTC alone. And they will be directed to the concerned people.”

#### 3.3.3. Supporting ADC Team and Communication

Effective communication and collaboration among different departments are essential for the successful implementation and operation of ADCs. This communication is particularly important for troubleshooting. All participants highlighted the need for efficient troubleshooting processes. Participant 3 emphasised the efficiency of communication between the vendor, pharmacy staff, biomedical technicians, nursing staff, and MID teams to address system issues and optimise performance. He said:

“When we have a system downtime, it’s easy to reach out to the team, reaching out to everyone from MID, nursing, pharmacy, and biomedical by phone is not difficult.”

Participant 7 explained the troubleshooting process, noting that the pharmacy is the first point of contact:

“We will call immediately the pharmacy. The pharmacy here will guide us.”

However, Participant 2 identified the heavy workload as a major barrier to effective communication and troubleshooting:

“It takes time because every department has its tasks, and everyone is busy here. Everyone is busy.”

Participant 5 agreed, stating:

“Sometimes the biomedical staff is busy, they will not come at the same time we need them. So, I keep calling them.”

One of the major issues in communication is medication verification by the pharmacy. Participants noted that response times were often longer than ideal. Participant 3 explained:

“All non-overridable medication needs to be verified by a pharmacist. The pharmacy staff is fine with that and dedicates a lot of time to that work. So, for the KPI, if you check the KPI, maybe it takes almost an average of 8–9 min. But sometimes, a medication may require 15 min to be verified. So, it depends on the peak. … I think 15 min is too long. You know the pharmacy will advocate for their side and the nursing will advocate for their side. It might be better to compromise and reduce the time to around 7 or 8 min.”

Participant 6 noted that communication issues often arise with the pharmacy team, whereas the biomedical team addresses technical problems promptly. Participant 6 said:

“As for the biomedical team, we contact them for printer issues and others, and they are solving them, they are solving a lot of issues. But our issue with the pharmacy team, the pharmacy staff always rely on the system balance, while we are the ones on the front lines, and here when we tell them, that they don’t have this medication available and the ADC is supposed to respond directly, but instead the pharmacy staff responds: ‘No… on the system there are like 2 or 3 available medications or something like this’. Right?”

In contrast, Participant 9 expressed satisfaction with the response level during troubleshooting:

“I have never faced a problem that I needed to contact them, and they did not answer. Never.”

Participants also advocated for improving the dedicated ADC team to address troubleshooting and enhance overall system performance. Participant 7 said:

“It is better to have a team, a specific team for ADC called ADC team. Maybe one from pharmacy, one from biomedical, one from IT. So, they can solve any issue in case of troubleshooting on the units.”

## 4. Discussion

The thematic analysis of the interview transcripts revealed the key benefits, barriers, and improvement opportunities for implementing ADC systems. The implementation of ADCs generally has positive outcomes for all staff. The system has brought about greater time efficiency and reduced workload, reduced medication errors, and enhanced medication tracking. However, there are barriers to their implementation, including changes in workflow and workload distribution, issues with cabinet design, technical medication management challenges, and the need for staff training. To optimise the use of ADCs, several improvement opportunities exist, such as improving the process of stock adjustment, streamlining discharge medications, avoiding double medication preparation, generating department-specific error notifications, and establishing ADT teams for better communication. Addressing these barriers and improvement areas can maximise the effectiveness of ADCs in improving patient care.

One of the primary benefits of ADCs is their ability to enhance time efficiency and reduce workload, allowing healthcare staff to focus more on patient care. The shift in workload distribution following the introduction of ADCs was a notable outcome. Initially, some nurses might be unfamiliar with the use of the system, but it is then recognised as very helpful for saving time and simplifying tasks. One of the participants particularly noted that it eliminates the need for nurses to wait 30 min to an hour for the pharmacy to dispense medications from the ADC, thus speeding up the process of administering medication. This advantage is especially vital in emergency cases and intensive care units, where the reliance on a floor stock system for frequent dose adjustments and quick access underscores a critical issue that ADCs can effectively resolve. The use of ADCs ensures timely medication administration in these urgent settings, beyond merely enhancing patient satisfaction [17,18]. The results from previous studies align closely with these findings, confirming that ADCs significantly enhance nurse satisfaction by reducing their workload and making their work easier. This is further evidenced by the fact the reduction in time spent on dispensing and preparing medications, which decreased by an average of 32 min per 8-h shift. This reduction in time allows nurses to allocate more of their shift to direct patient care activities, reflecting a similar outcome observed in this study, where nurses experienced increased efficiency and were able to dedicate more time to patient-centered tasks [14,19,20]. These consistent findings demonstrate that ADCs not only streamline medication management but also contribute to improved nurse and patient satisfaction along with enhanced workload management.

Nurse managers also noted a decrease in their workload, in which ADCs cut down the time previously spent requesting medications for each department. Before the implementation of ADCs, nurse managers would spend up to two hours each night shift managing medication requests. In line with these observations, earlier research affirms these benefits [17,18,21,22]. Additionally, a notable point is that ADC was reported to save considerable nursing time in acquiring controlled substances and managing their inventory during shift changes [22,23]. This may explain why nurses have more time now with the use of ADC, given that these systems eliminate the need for manual narcotic counts at the end of shifts in patient care units [17,24].

In addition, ADC offers significant benefits to pharmacists by streamlining medication preparation and distribution processes. One participant specifically noted that the system alleviated significant work for pharmacists, in which it was previously time-consuming to prepare medications and wait for trollies to be brought to them. Although it was reported that ADCs have improved the workflows and reduced the challenges faced by both pharmacists and nurses, one participant who is a pharmacist, pointed to a different perspective. The participant initially thought that the machine would significantly reduce their workload. However, for pharmacists, the workflow was primarily shifted with increased responsibilities given to those stocking the cabinets, while the workload for other pharmacists decreased. In contrast to the literature, where pharmacy technicians are typically solely responsible for stocking, the duties of stocking and related responsibilities, in our institution, this task is distributed among pharmacists and alternated between shifts. This explains participants’ statements that with the integration of ADCs, inpatient pharmacists have experienced a reduced workload compared to pharmacists who are responsible for ADC stocking and managing. These are consistent with findings from other studies on the impact of ADCs, where pharmacy technicians who handled stock checks and stocking machines had more workload and spent more time, while pharmacists had a reduction in medication supply workload [15,25,26]. Therefore, these changes in the workflow and workload distribution are both benefits and barriers at the same time. The alterations in workflow and the redistribution of workload present both a significant benefit and a potential hindrance simultaneously.

Furthermore, the implementation of ADCs offers a critical benefit in reducing errors and improving patient safety. As noted by Participant 6, ADCs improve error prevention by displaying exact dosages and alerting staff when the dispensed amount exceeds the required dose. However, despite these measures, human error can still occur, especially among less experienced nurses, as they may overlook the instructions shown in the ADC. Nonetheless, it is well-established that the integration of ADCs fundamentally improves the accuracy of medication administration and optimises both patient safety and hospital efficiency [3,6,7,13]. In our institution, all medication prescriptions must be approved by a pharmacist before nurses can retrieve medications from ADCs. The override function is limited to specific medications and is used only in emergencies, which further emphasises the institution’s dedication to ensuring patient safety. By limiting overrides to critical situations, the risk of bypassing essential safety checks, which could contribute to medication errors, has been minimised. This approach is recommended by the Emergency Care Research Institute (ECRI) and supported by the literature [17,27,28,29].

The technology also supports enhanced medication tracking, which ensures better accountability and control over medication use. The participants highlighted that ADCs play a crucial role in monitoring medication dispensing and stock levels, a significant advantage that is supported by the existing literature [25,26]. Participant 3 emphasised that while trust in nursing staff is essential, having a system in place to track and ensure accuracy is invaluable and that the precise tracking of medication stocks also eliminates the need for additional audits leading to a more self-sufficient system. Participant 6 also emphasised the importance of having medication dispensing reports available on their system, which facilitates better tracking and management.

Despite the significant and valuable advantages of ADC, several barriers to the effective use of these systems emerged. A common issue reported by the participants was staff training. The introduction of a new medication dispensing system may pose potential risks to medication safety. To mitigate these risks, comprehensive staff training and strategic reallocation of resources are required [13]. These proactive measures are critical to ensuring a seamless transition and maintaining high standards of patient safety throughout the implementation process. During the interviews, the participants emphasised that training was a huge obstacle in the ADC integration process, as Participant 1 noted that the training process for both the pharmacy team and the nurses is huge work. Similarly, Participant 4 stressed that without the training, the staff would not be able to use the ADC, highlighting the essential role of personnel training in the integration process. The complexity of adapting to new systems demanded rigorous training for both pharmacy teams and nurses. Participants also noted that personnel training must be customised and ongoing training and monitoring to meet individual needs and ensure the safe and effective use of ADCs. Participants’ emphasis on the training process as a significant barrier highlights the crucial need for thorough preparation. This focus highlights the critical need for detailed and comprehensive training, as emphasised in the literature. Such training is essential for the successful integration of new technologies and for maintaining high standards of patient and staff safety [4,30,31].

Moreover, the study revealed that technical issues were significant barriers related to the implementation and operation of ADCs. Specifically, integration problems with existing systems, such as ERP, network, and downtime issues have emerged as notable challenges impacting the integration of ADCs. The experience of Participant 2 highlights a critical issue with system integration. Participant 2 reported a problem where medication refills were not recorded in the system due to integration issues with ERP, leading to significant disruptions in medication management. When the system fails to update or reflect current medication refills, it not only hinders medication availability but also poses risks to patient safety. During the interviews, the participants also reported charging discrepancies during ADC, especially during downtimes. Participants provided contrasting perspectives on the source of the problem. Participant 4 highlighted a technical issue of medication double charging, which usually occurs during downtimes. This observation points to a potential limitation in the ADC system’s reliability. In contrast, Participant 3 emphasised that the root cause of charging issues is often human behaviour rather than technical malfunctions. According to this participant, improper staff training and a lack of oversight contribute more to these discrepancies than system errors. These insights underscore the importance of staff education and monitoring as critical components of effective ADC management. Both perspectives highlight different but complementary strategies for mitigating charging inconsistencies, suggesting that technical and human factors must be considered in optimising ADC performance. These technical issues underscore the critical role of shared responsibility in providing support and troubleshooting from IT and informatics teams. Effective IT support is essential for resolving integration problems swiftly and ensuring system stability [32]. Additionally, the observation of Participant 4 about network and integration issues, particularly during system downtime, further emphasise the broader scope of technical challenges faced by ADCs. Network instability can cause delays or interruptions in accessing medication, which may impact staff efficiency and patient care. Therefore, maintaining a resilient network infrastructure is crucial to support the uninterrupted functioning of ADC systems [33,34]. The technical challenges encountered in this study reflect broader issues reported in the literature for ADCs and other healthcare technologies [9,32,33,35].

Regarding the open free zones, some participants viewed them as beneficial, while others had concerns in terms of selecting the right medication. Participant 2 highlighted the benefits of these features for accommodating larger vials. In contrast, Participant 3, Participant 4, and Participant 5 expressed safety concerns, preferring fully secured cabinets. During the interviews, Participant 5 highlighted a critical concern regarding the free zone design in the ADC. The close proximity of free zones can lead to self-doubt and selecting the wrong free zone. Thereby raises the risk of medication errors, as staff may inadvertently select the wrong medication, particularly when under a high workload. Let alone that when nurses doubt themselves, this can potentially increase stress and reduce overall efficiency. This issue emphasises the importance of clear designs or better separation between medication compartments to ensure accurate medication selection. Moreover, another critical challenge that emerged with the use of ADCs is the management of stock adjustment and the tracking of medication discrepancies. With the implementation of ADC systems, nurses are now required to perform regular stock adjustments [36,37]. The participants stated that the system stock adjustment is a heavy task and time-consuming, and sometimes they are not able to keep up with the demands of checking stock adjustment during high-volume times. Although the participants in this study expressed that system stock adjustment is challenging and time-consuming, previous studies have reported opposite experiences regarding the narcotic daily count. In those studies, the reduction in end-of-shift narcotic count by nurses was considered a significant advantage of implementing ADCs [24,27,38].

Concerning the improvement opportunities, the participants highlighted several critical areas where ADCs could be improved to enhance operations and staff efficiency. One major suggestion was having error notification reports, specifically identified with the respective department, which would allow for faster issue resolution. Through better identification and classification of error notification reports, the process becomes more focused and efficient. Another key area for improvement was the inclusion of discharge medications in the ADCs to reduce delays in patient discharges. One participant reported that in the NICU, where discharge medications are often limited to items, delays occur when these medications are not readily available in the system. Enabling the charging of discharge medications in ADCs would help expedite the process. Participants also recommended incorporating features that prevent double medication preparation for already dispensed medication, as well as emergent medication requests. In such cases, nurses might dispense medications that have already been dispensed and administered or which are usually prepared by the inpatient intravenous (IV) pharmacy. Not only does double medication preparation lead to unnecessary costs, but it also raises safety concerns when medications are dispensed twice. Suggestions included locking the ADC after a medication has been dispensed, preventing multiple dispensing of the same order. In addition, enhancing clear communication and teamwork among various departments is a critical area for improvement, as it plays a vital role in the successful implementation and operation of ADCs. Participants emphasised the crucial need for creating a specialised ADC team and reallocating tasks, such as stock adjustment, to the pharmacy, which would reduce the burden on other departments. They suggested that this dedicated ADC team should adopt a multidisciplinary approach, involving pharmacy, biomedical, and MID staff. Previous studies have emphasised the significance of forming a specific team for ADC during the initiation period. However, they often do not address the need for ongoing communication and support beyond this initial phase [21,39]. This highlights a critical issue and stresses the need for sustained team engagement and communication.

This study uncovered critical insights that establish a robust framework for understanding the implementation of ADCs. The findings powerfully highlight important aspects such as benefits, barriers, and transformative opportunities for improvement, significantly enriching the body of knowledge in this field. However, a limitation of this study is the variation between different manufacturers of ADCs, which may impact the generalisability of the findings. Additionally, when participants were asked about the barriers and challenges, they may have focused on rare incidents as significant issues, potentially generalising these specific cases as broader, common challenges. This could lead to an altered understanding of the nature and widespread occurrence of the challenges associated with ADCs across various contexts.

## 5. Conclusions

The implementation of ADCs offers significant potential to enhance healthcare delivery by improving time efficiency, reducing medication errors, and enabling more effective medication tracking. Despite the numerous benefits these technologies offer, there are potential challenges encountered throughout the process, such as the need for improved staff training, workflow adjustments, and the resolution of technical and design-related issues. Improvement opportunities also exist, which can further optimise the use of ADCs. These include establishing consistent effective communication across departments, ensuring the avoidance of double medication preparation, and tailoring reports specific to each department to address unique challenges. To optimise the use of ADCs, several improvement opportunities exist, including improving stock adjustment processes and supporting ADT teams for better communication between various departments. Further attention should be given by manufacturers of ADCs to focus on technological advancements, which include improving discharge medication processes and generating department-specific error notifications. Additionally, healthcare organisations should prioritise continuous staff training programs to ensure that all personnel are proficient in operating ADCs to their fullest potential. In parallel, departments must actively monitor both the technical and operational performance of these systems. This enables adjustments to system configurations, ensuring that the technology evolves in line with organisational needs and operational challenges. By integrating comprehensive training with continuous performance monitoring, healthcare organisations can maximise the benefits of ADCs while addressing any emerging technical or design-related issues.

## Figures and Tables

**Figure 1 pharmacy-13-00012-f001:**
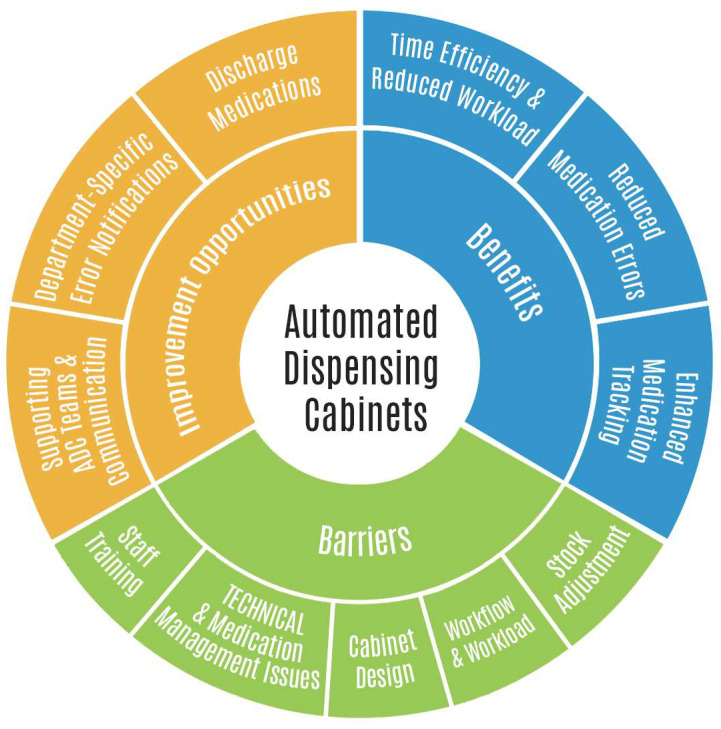
Implementing Automated Dispensing Cabinets.

**Table 1 pharmacy-13-00012-t001:** Summary of the participant’s characteristics.

Participant	Sex	Job	Department	Work Experience	Experience with ADCs
Participant 1	Female	Medical Informatics Pharmacist	Medical Informatics Department	18 years	Her journey with ADC began in 2021 with a pilot phase preceded by nine months of integration efforts between ADC and the hospital’s HIS.
Participant 2	Female	Pharmacist	Pharmacy Department	2.5 years	She has been working with the ADC as an admin for nearly two years.
Participant 3	Male	Nurse Manager	Emergency Room (ER)	22 years	He has overseen the implementation of the ADC.
Participant 4	Female	Charge Nurse	Emergency Room (ER)	20 years	She has been working with the ADC since it was implemented in the hospital.
Participant 5	Female	Nurse Manager	Neonatal Intensive Care Unit (ICU)	24 years	She has been working with the ADC since it was implemented in the hospital
Participant 6	Male	Nurse Manager	Adult ICU	20 years	The ADC system was introduced at the hospital shortly before he joined the hospital.
Participant 7	Female	Nurse Manager	Bariatric Surgery Unit	8 years	She used a different type of ADC in her previous role outside of Saudi Arabia.
Participant 8	Male	Nurse Manager	Orthopedics Unit	24 years	He has been working with ADC since it was implemented in the hospital.
Participant 9	Female	Nurse Manager	Oncology	7.5 years	She has been working with the ADC system for nearly two years.
Participant 10	Female	Nurse Manager	General Medical Unit	15 years	She has been working with ADC for nearly two years.

ADC: Automated Dispensing Cabinet, HIS: Healthcare Information System.

**Table 2 pharmacy-13-00012-t002:** Summary of the findings on the Automated Dispensing Cabinet.

Theme	Subtheme	Codes	Excerpt
Benefits	Time Efficiency and Reduced Workload	Time Saving Reduced Workload for Nurses Automation of Medication Requests Simplified Processes	“The hospital work for nurses has decreased. Before, when we had only floor stock medications, I would spend two hours per day when it is the night shift to request medications. You will count how many were left and you will request, it will take time, it will take two hours, now no more.” (Participant 4)
	Reduced Medication Errors	Regulated Stock and Accurate Dosing Error Prevention Features Safe Practice Maintenance Sufficient Reporting Error Notifications and Tracking Human Error Risk	“Working with ADC really maintains a safe practice and reduces medication errors.” (Participant 6)
	Enhanced Medication Tracking	Tracking Staff Behavior Reports and Sufficient Notifications	“Honestly, the pharmacists are doing a great job, every day the pharmacists send us the report of the stock discrepancies and we use this report to correct them immediately. So, we do not need to go for a stock adjustment every time.” (Participant 3)
Barriers	Staff Training	Training Importance Training Duration and Frequently Vendor-led Training	“For each process, you need training. For a new process that will be introduced to any organization, you must follow the guidelines communicated during the training. Without the training, you cannot do it.” (Participant 4)
	Technical and Medication Management Issues	System Integration Issues Medication Charging Inaccuracies System Downtime Overridable Medications Risk Human Behavior	“Network and integration issues, especially during downtime.” (Participant 4)
	Cabinet Design	Free Zone and Light Confusion Staff Compliance	“In the beginning, it was a little confusing for the light because the light or the drawer, when it’s opened, will show you the light itself and this light, it’s coming between two drawers.” (Paticipant 6)
	Workflow and Workload Distribution	Refilling Process Administrative Tasks	“The refilling process is hard, it’s not easy. So, we are taking almost one hour… one hour and a half every morning to check the medication we need to refill…” (Participant 2)
	Stock adjustment	Medication Discrepancies Medication Counting	“The system stock adjustment, here I have an issue, I mean, I don’t always have time to do a system stock adjustment.” (Participant 6)
Improvement Opportunities	Discharge Medications	Discharge Delay	“In NICU, the discharge medication is only vitamin D, we rarely have another medication. Sometimes it will cause a delay of discharge because we are waiting for vitamin D. And if we can add the discharge medication and make it chargeable.” (Participant 5)
	Department-Specific Error Notifications	Error Notification by Department	“If we can identify it for emergency or a specific department, it will be much better.” (Participant 3)
	Supporting ADC Team and Communication	Communication Faster Response Times ADC Teems	“I have never faced a problem that I needed to contact them, and they did not answer. Never.” (Participant 9)

## Data Availability

The data presented in this study are available on request from the corresponding author.

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
