# Peer review of "Exploring the Benefits, Barriers and Improvement Opportunities in Implementing Automated Dispensing Cabinets: A Qualitative Study"

_pharmacy, 2025, doi:10.3390/pharmacy13010012_

Round 1
Reviewer 1 Report
Comments and Suggestions for Authors
In the introduction to this study, it was felt necessary to explain in detail the organisation and functioning of the health system in Saudi Arabia: i.e. to understand the process from the time a user visits a health professional until he/she is able to obtain the medicines prescribed by the health professional.
For example, it is not clear whether the nursing staff or the person in charge of the pharmacy (pharmacist) can prescribe medicines or only a medical practitioner (doctor) can prescribe them.
It is also necessary to state at the beginning of the work that only the opinion of the professionals will be sought, and, in the same vein, to explain why there are only nursing and administrative staff, but no medical staff.
It may also be surprising that the opinion of users is not taken into account and the reason for this absence is not explained.
It would also be necessary to have a little more detail on how the interviews were conducted: sampling process, location of the interviews, duration of the interviews, interviewers present in each of them, etc.
In addition, the numbering of the table presented does not match the numbering of the presentation of the results.
Finally, it is believed that the presentation of the results should be improved, as the reading of all the opinions of the participants without grouping them by thematic blocks or in a summary table is a bit monotonous and not very clarifying.
Author Response
Reviewer #1:
- In the introduction to this study, it was felt necessary to explain in detail the organisation and functioning of the health system in Saudi Arabia: i.e. to understand the process from the time a user visits a health professional until he/she is able to obtain the medicines prescribed by the health professional. For example, it is not clear whether the nursing staff or the person in charge of the pharmacy (pharmacist) can prescribe medicines or only a medical practitioner (doctor) can prescribe them.
Noted and a new paragraph was added about pharmacy practice in Saudi Arabia. The paragraph was solely focused on in-patient dispensing, as the study is about. Regarding the responsibility of prescribing in Saudi Arabia, it varies greatly between institutions. There is a challenge in the recognition of nurse practitioners and defining their scope of practice in Saud Arabia, and thus, only few institutions (approx. 3) have implemented this and allowed them to prescribe medications. General pharmacists are not allowed to prescribe, unlike clinical pharmacists (Doctor of Pharmacy) and they work closely with physicians, and this is the practice in our institution. Regarding the ADC overriding orders, nurses can override for a specific medications only.
- It is also necessary to state at the beginning of the work that only the opinion of the professionals will be sought, and, in the same vein, to explain why there are only nursing and administrative staff, but no medical staff. It may also be surprising that the opinion of users is not taken into account and the reason for this absence is not explained.
We intended to involve all stakeholders and managers, as they are being involved deeply in the implementation process, knowing the concerns and issues raised by their staff within each department.
- It would also be necessary to have a little more detail on how the interviews were conducted: sampling process, location of the interviews, duration of the interviews, interviewers present in each of them, etc.
“The interviews were held in a meeting room at the hospital. Each interview took about 30 to 40 minutes. All interviews were attended by two investigators, A.A.M. and K.T. The interviewers were and the research center director and the pharmacy supervisor. The interview guide was prepared ahead of time and were the same for all interviews. The questions were open-ended to allow detailed answers.”
- In addition, the numbering of the table presented does not match the numbering of the presentation of the results.
Our apologies, information within the cells got mixed-up as we were transferring them to the journal template. We revised this issue. Thank you for your comment and consideration.
- Finally, it is believed that the presentation of the results should be improved, as the reading of all the opinions of the participants without grouping them by thematic blocks or in a summary table is a bit monotonous and not very clarifying.
Noted and revised accordingly. Please refer to Table 2.

Reviewer 2 Report
Comments and Suggestions for Authors
This qualitative exploration of ADCs involves staff interviews post ADC implementation. Automation is a current topic of interest. While technology is to be welcomed, of equal importancce is reviewing the technology post implementation. The views of the participants is of interest . However the paper would benefit from streamlining
Introduction:
line 46 reducing medication errors and preventing delays - does this refer to the institution or in general? Statement s are not referenced.
When were ADCs introduced in this institution?
Describe briefly the method of medication distribution prior to ADC introduction
line 56- to what does 8.4% refer?
Methods
What were the specific questions asked of the participants?
How were staff invited to participate? that is, was the 'purposive sampling' done to minimise bias
Please indicate also which authors belong to which Department/clinical specialty (Note- rather than "job")
Table 1- participants are numbered 1 to 7 and then 1 to 4.. Why not 1 to 11 ? Option of not having this table and just stating how many staff were surveyed, their professions and mean years of experience in Methods
Results
There are too many quotes . I suggest major themes, with an illustrative quote for each, referring to the participant in brackets eg " I hate ADCs" ( Pharmacist, 2 years' experience) ; " My workload is decreased" ( Nurse 22 years experience)
3.1.2 and 3.1.3 are both "Reduced medication errors"
3.2.3 Ensure terms are explained eg 'free zone' and 3.2.5 "system stock adjustment"
Please be consistent - suggest using 'medication' rather than 'drug' where possible e.g. line 467
Consider highlighting positives and negatives in a Table, as this is lost in the text
Discussion
This should be a short summary of the main findings- a lot is repeated from the Result and Introduction sections. What did this study find that is different to work already published? What were the 'valuble insights"mentioned in the introduction to assist others in implementing ADCs?
line 678, 679 'critical' repeated.
line681- the limitation of the study isn't the variety of ADCs manufactured- that is a issue of generalisability.
Author Response
Reviewer #2:
This qualitative exploration of ADCs involves staff interviews post ADC implementation. Automation is a current topic of interest. While technology is to be welcomed, of equal importancce is reviewing the technology post implementation. The views of the participants is of interest . However the paper would benefit from streamlining
Introduction:
- line 46 reducing medication errors and preventing delays - does this refer to the institution or in general? Statement s are not referenced.
Noted and revised accordingly.
- When were ADCs introduced in this institution?
Noted and revised accordingly.
“The integration of ADCs within XXX was introduced in September 2021, as part of an initiative to enhance medication management efficiency and patient safety. Currently, the hospital has deployed 25 functional ADCs across 20 units.”
- Describe briefly the method of medication distribution prior to ADC introduction
Noted and revised.
“Prior to ADC implementation, the hospital was using both manual floor stock systems and medication carts for storing a 24-hour medications supply with patient-specific containers.”
- line 56- to what does 8.4% refer?
Noted and revised accordingly added “of healthcare professionals have previously”.
“8.4% of healthcare professionals have previously reported that the use of ADCs is time-consuming and more difficult”
Methods
- What were the specific questions asked of the participants?
Please refer to the attached Interview Guide.
- How were staff invited to participate? that is, was the 'purposive sampling' done to minimise bias
Purposive sampling was utilized to ensure the involvement of key stakeholders in the implementation process, including pharmacy staff, medical informatics pharmacist and nurse managers. The staff participated in the interview were chosen based on their roles, with each being responsible for a department involved in the implementation.
We intended to involve all stakeholders and managers, as they are being involved deeply in the implementation process, knowing the concerns and issues raised by their staff within each department.
- Please indicate also which authors belong to which Department/clinical specialty (Note- rather than "job")
Noted and revised: “The interviewers were and the research center director and the pharmacy supervisor.”
As to the rest of the authors: K.A.: research specialist, B.A. Pharmacy Director. A.E.: Dentist, C.S: research specialist. H.A. and M.A.: pharmacist.
- Table 1- participants are numbered 1 to 7 and then 1 to 4.. Why not 1 to 11 ? Option of not having this table and just stating how many staff were surveyed, their professions and mean years of experience in Methods
Our apologies, information within the cells got mixed-up as we were transferring them to the journal template. We revised this issue. Thank you for your comment and consideration.
Results
There are too many quotes . I suggest major themes, with an illustrative quote for each, referring to the participant in brackets eg " I hate ADCs" ( Pharmacist, 2 years' experience) ; " My workload is decreased" ( Nurse 22 years experience)
- 1.2 and 3.1.3 are both "Reduced medication errors"
Noted and revised. (3.1.2 is Reduced medication errors, 3.1.3 is Enhanced Medication Tracking).
- 2.3 Ensure terms are explained eg 'free zone' and 3.2.5 "system stock adjustment"
Noted and revised.
- Please be consistent - suggest using 'medication' rather than 'drug' where possible e.g. line 467
Noted and revised.
- Consider highlighting positives and negatives in a Table, as this is lost in the text
Noted and revised. Please refer to Table 2.

Reviewer 3 Report
Comments and Suggestions for Authors
I read with interest the paper titled "Exploring the Benefits, Barriers and Improvement Opportunities in Implementing Automated Dispensing Cabinets: A Qualitative Study"
1. Afiliation should be reviewed as per journal rules.
2. What's the advantages of ADC vs the traditional method? This should be extensively discussed in the background.
3. "Sex" should be used instead of "gender"
4. Why table 1 have multiple times the same participants? How many participants were engaged? This should be described.
5. participant 10 is appearing in the description of statements, but I didn't found participant 10 in Table 1. The same for participant 8 and participant 9.
6. Authors stated that 4 themes are in Figure 1. I dont find the themes there. If themes are benefits, barriers and improved benefits, they are 3, not 4.
7. What's the software used for thematic analysis?
8. Please provide more quotes to each sub-theme.
9. Please provide as appendix the list of all quotes coded, by theme and subtheme.
Author Response
Reviewer #3:
I read with interest the paper titled "Exploring the Benefits, Barriers and Improvement Opportunities in Implementing Automated Dispensing Cabinets: A Qualitative Study"
- Affiliations should be reviewed as per journal rules.
- What's the advantages of ADC vs the traditional method? This should be extensively discussed in the background.
Noted and revised.
- "Sex" should be used instead of "gender"
Noted and revised.
- Why table 1 have multiple times the same participants? How many participants were engaged? This should be described. 5. participant 10 is appearing in the description of statements, but I didn't found participant 10 in Table 1. The same for participant 8 and participant 9.
Our apologies, information within the cells got mixed-up as we were transferring them to the journal template. We revised this issue. Thank you for your comment and consideration.
- Authors stated that 4 themes are in Figure 1. I dont find the themes there. If themes are benefits, barriers and improved benefits, they are 3, not 4.
Noted and revised.
- What's the software used for thematic analysis? Please provide more quotes to each sub-theme. Please provide as appendix the list of all quotes coded, by theme and subtheme. Microsoft Excel
Noted and revised, please refer to the manuscript, Table 2.

Round 2
Reviewer 2 Report
Comments and Suggestions for Authors
Thanksyou for addressing the reviewer comments
Reviewer 3 Report
Comments and Suggestions for Authors
The authors adressed my previous questions. I have no further comments to add.